# ENFORCING FAIRNESS IN PRIVATE FEDERATED LEARNING VIA THE MODIFIED METHOD OF DIFFERENTIAL MULTIPLIERS

## ABSTRACT

Federated learning with differential privacy, or private federated learning, provides a strategy to train machine learning models while respecting users' privacy. However, differential privacy can disproportionately degrade the performance of the models on under-represented groups, as these parts of the distribution are difficult to learn in the presence of noise. Existing approaches for enforcing fairness in machine learning models have considered the centralized setting, in which the algorithm has access to the users' data. This paper introduces an algorithm to enforce group fairness in private federated learning, where users' data does not leave their devices. First, the paper extends the modified method of differential multipliers to empirical risk minimization with fairness constraints, thus providing an algorithm to enforce fairness in the central setting. Then, this algorithm is extended to the private federated learning setting. The proposed algorithm, FPFL, is tested on a federated version of the Adult dataset and an "unfair" version of the FEMNIST dataset. The experiments on these datasets show how private federated learning accentuates unfairness in the trained models, and how FPFL is able to mitigate such unfairness.

## 1 INTRODUCTION

Machine learning requires data to build models. This data is often private and resides on users' devices. *Federated learning* (FL) (McMahan et al., 2017) is a strategy where multiple users (or other entities) collaborate in training a model under the coordination of a central server or service provider. In FL, devices share only data statistics (e.g. gradients of the model computed from their data) with the server, and therefore the users' data never leaves their devices.

However, these statistics can still leak information about the users (see Shokri et al. (2017); Fredrikson et al. (2015) for practical attacks recovering the users' identity and reconstructing the users' face images). To protect against this leakage, *differential privacy* (DP) can be provided. Differential privacy offers a mathematical guarantee on the maximal amount of information any attacker can obtain about a user after observing the released model. In practice, often (Bhowmick et al., 2018; McMahan et al., 2018a;b; Truex et al., 2019; Granqvist et al., 2020) this guarantee is achieved by adding a certain amount of noise to the individuals' statistics. In this paper, federated learning with differential privacy will be referred to as *private federated learning* (PFL).

The models trained with PFL are often neural networks, since they work well in many use cases, and turn out to be resistant to the noise added for DP. These models have been successfully trained with PFL for tasks like next word prediction or speaker verification (McMahan et al., 2017; Granqvist et al., 2020). However, they are prone to perpetuating societal biases existing in the data (Caliskan et al., 2017) or to discriminate against certain groups even when the data is balanced (Buolamwini & Gebru, 2018). Moreover, when the training is differentially private, degradation in the performance of these models disproportionately impacts under-represented groups (Bagdasaryan et al., 2019). More specifically, the accuracy of the model on minority groups is deteriorated to a larger extent than the accuracy for the majority groups.

In the realm of FL, there has been some research studying how to achieve that the performance of the model is similar among devices (Hu et al., 2020; Huang et al., 2020; Li et al., 2019; 2021). However,

this notion of fairness falls short in terms of protecting users from under-represented groups falling foul of disproportionate treatment. For example, Castelnovo et al. (2021, Section 3.6) show that a model that performs well on the majority of the users will have a good score on individual fairness metrics, even if all the users suffering from bad performance belong to the same group. Conversely, there is little work proposing solutions to enforce group fairness, i.e. that the performance is similar among users from different groups. Current work in this area is either limited to reducing demographic parity in logistic regression (Du et al., 2021) or to minimizing the largest loss across the different groups with a minimax optimization (Mohri et al., 2019). Concurrently to this paper, Cui et al. (2021) proposed an approach to enforce group fairness in FL for general models and metrics. However, this approach does not consider DP. Moreover, most prior work does not consider the adverse effects that DP has on the models trained with private federated learning, or is restricted to logistic regression models (Abay et al., 2020).

On the other hand, work studying the trade-offs between privacy and group fairness proposes solutions that are either limited to simple models such as linear logistic regression (Ding et al., 2020), require an oracle (Cummings et al., 2019), scale poorly for large hypothesis classes (Jagielski et al., 2019), or only offer privacy protection for the variable determining the group (Tran et al., 2021; Rodríguez-Gálvez et al., 2021). Furthermore, the aforementioned work only considers the central learning paradigm, and the techniques are not directly adaptable to FL.

For all the above reasons, in this paper, we propose an algorithm to train neural networks with PFL while enforcing group fairness. We pose the problem as an optimization problem with fairness constraints and extend the modified method of differential multipliers (MMDM) (Platt & Barr, 1987) to solve such a problem with FL and DP. Hence, the resulting algorithm (i) is applicable to any model that can be learned using stochastic gradient descent (SGD) or any of its variants, (ii) can be tailored to enforce the majority of the group fairness metrics, (iii) can consider any number of attributes determining the groups, and (iv) can consider both classification and regression tasks.

The paper is structured as follows: in Section 2, we review the background on differential privacy, private federated learning, the MMDM algorithm, and group fairness; in Section 3, we present our approach for fair private federated learning; in Section 4, we describe our experimental results; and in Section 5, we conclude with a summary and interpretation of our findings.

## 2 BACKGROUND

In this section, we review several aspects on the theory of private federated learning, the MMDM algorithm, and group fairness that are necessary to develop and understand the proposed algorithm.

### 2.1 FEDERATED LEARNING

The federated learning setting focuses on learning a model that minimizes the expected value of a loss function $\ell$ using a dataset $d = (\boldsymbol{z}_1, \boldsymbol{z}_2, \ldots, \boldsymbol{z}_n) \in \mathcal{D}$ of $n$ samples distributed across $K$ users. This paper will consider models parametrized by a fixed number of parameters $\boldsymbol{w} \in \mathbb{R}^p$ and differentiable loss functions $\ell$, which includes neural networks.

Since the distribution of the data samples $Z$ is unknown and each user $k$ is the only owner of their private dataset $d^k$, the goal of FL is to find the parameters $\boldsymbol{w}^\star$ that minimize the loss function across the samples of the users. That is, defining $L(d^k, \boldsymbol{w}) := \sum_{\boldsymbol{z} \in d^k} \ell(\boldsymbol{z}, \boldsymbol{w})$, to solve

$$\boldsymbol{w}^\star = \arg\min_{\boldsymbol{w} \in \mathbb{R}^p} \frac{1}{n} \sum\nolimits_{k=1}^{K} L(d^k, \boldsymbol{w}). \tag{1}$$

This way, the parameters $\boldsymbol{w}$ can be learned with an approximation of SGD. McMahan et al. (2017) suggest that the central server iteratively samples $m$ users; they compute and send back to the server an approximation of the model's gradient $\nabla_{\boldsymbol{w}} L(d^k, \boldsymbol{w})$; and finally the server updates the parameters as dictated by gradient descent $\boldsymbol{w} \leftarrow \boldsymbol{w} - \frac{1}{n} \sum_k \eta \nabla_{\boldsymbol{w}} L(d^k, \boldsymbol{w})$, where $\eta$ is the learning rate. If the users send back the exact gradient the algorithm is known as `FederatedSGD`. A generalisation of this, `FederatedAveraging`, involves users running several local epochs of mini-batch SGD and sending up the difference. However, the method in this paper considers an optimization with fairness constraints, requiring the communication of extra information in each iteration, and therefore will extend `FederatedSGD`.

## 2.2 Private federated learning

Private federated learning combines federated learning with differential privacy. Differential privacy (Dwork et al., 2006; 2014) provides a mathematical guarantee by restricting the amount of information that a randomized function $m$ of a dataset $d$ leaks about any individual. More precisely, it ensures that the probability that the output of the function $m$ on the dataset $d$ belongs to any set of outcomes $\mathbb{O}$ is close to the probability that the output of the function $m$ applied to a neighbouring dataset $d'$ (i.e. equal to $d$ but where all samples from any individual are removed or replaced) belongs to said set of outcomes. Formally, a randomized function $m$ is $(\epsilon, \delta)$-differentially private if

$$\mathbb{P}\big[m(d) \in \mathbb{O}\big] \leq e^\epsilon \mathbb{P}\big[m(d') \in \mathbb{O}\big] + \delta, \tag{2}$$

where $\epsilon$ and $\delta$ are parameters that determine the strength of the guarantee. Two useful properties of DP are that that it composes (i.e. the combination of DP functions is DP) and that post-processing does not undo its privacy guarantees (Dwork et al., 2014). These properties ensure that the modifications that we will make to `FederatedSGD` will keep the algorithm differentially private.

Typically, the functions of interest are not differentially private. For this reason, they are privatized with "privacy mechanisms", which apply a random transformation to the output of the function on the dataset. A standard mechanism is the Gaussian mechanism, which adds Gaussian noise to the function's output, calibrating the noise variance with the privacy parameters $(\epsilon, \delta)$ and the maximum individual's contribution to the function's output measured by the $\ell_2$ norm.

In the context of `FederatedSGD`, the functions to privatize are the gradient estimates sent to the server. The individual's contribution is the gradient computed on their data. This gradient is "norm-clipped" (Abadi et al., 2016), i.e. it is scaled down so its $\ell_2$ norm is at most a predetermined value $C$. Then, Gaussian noise is added to the sum of contributions, which is sent to the server.

Applying the privacy mechanism to a sum of contributions is a form of central DP. An alternative is local DP (Dwork et al., 2014), where the privacy mechanism is applied to the statistics before leaving the device. However, models trained with local DP often suffer from low utility (Granqvist et al., 2020), and are thus not considered in this paper. Central DP can involve trust in the server that updates the model, or a trusted third party separated from the model, which is how DP was originally formulated. This paper instead assumes the noisy sum is computed through secure multi-party computation (Goryczka & Xiong, 2015), such as secure aggregation (Bonawitz et al., 2017). This is a cryptographic method that ensures the server only sees the sum of contributions. A crucial limitation of secure aggregation is that it can only compute sums, so the novel method introduced in Section 3 is restricted to working only on sums of individuals' statistics.

## 2.3 The Modified Method of Differential Multipliers

Ultimately, our objective is to find a parametric model that minimizes a differentiable loss function and respects some fairness constraints. Thus, this subsection reviews a constrained differential optimization algorithm, the modified method of differential multipliers (MMDM) (Platt & Barr, 1987).

This algorithm tries to find a solution to the following constrained optimization problem:

$$\boldsymbol{w}^\star = \arg\min_{\boldsymbol{w} \in \mathbb{R}^p} \phi(\boldsymbol{w}) \quad \text{s.t. } \boldsymbol{g}(\boldsymbol{w}) = \boldsymbol{0}, \tag{P1}$$

where $\phi : \mathbb{R}^p \to \mathbb{R}$ is the function to minimize, $\boldsymbol{g}(\boldsymbol{w}) = (g_1(\boldsymbol{w}), g_2(\boldsymbol{w}), \ldots, g_r(\boldsymbol{w}))$ is the concatenation of $r$ constraint functions, and $\{\boldsymbol{w} \in \mathbb{R}^p : \boldsymbol{g}(\boldsymbol{w}) = 0\}$ is the solution subspace.

The algorithm consists of solving the following set of differential equations resulting from the Lagrangian of (P1) with an additional quadratic penalty $c\boldsymbol{g}(w)^2$ using gradient descent/ascent. This results in an algorithm that applies these updates iteratively:

$$\begin{cases} \boldsymbol{\lambda} \leftarrow \boldsymbol{\lambda} + \gamma \boldsymbol{g}(\boldsymbol{w}) \\ \boldsymbol{w} \leftarrow \boldsymbol{w} - \eta\Big(\nabla_{\boldsymbol{w}}\phi(\boldsymbol{w}) + \boldsymbol{\lambda}^T \nabla_{\boldsymbol{w}}\boldsymbol{g}(\boldsymbol{w}) + c\boldsymbol{g}(\boldsymbol{w})^T \nabla_{\boldsymbol{w}}\boldsymbol{g}(\boldsymbol{w})\Big) \end{cases}, \tag{3}$$

where $\boldsymbol{\lambda} \in \mathbb{R}^r$ is a Lagrange (or dual) multiplier, $c \in \mathbb{R}_+$ is a damping parameter, $\eta$ is the learning rate of the model parameters, and $\gamma$ is the learning rate of the Lagrange multiplier.

Intuitively, these updates gradually fulfill (P1). Updating the parameters $\nabla_{\boldsymbol{w}}\phi(\boldsymbol{w})$ and $\nabla_{\boldsymbol{w}}\boldsymbol{g}(\boldsymbol{w})$ respectively enforce function minimization and constraint's satisfaction. Then, the multiplier $\boldsymbol{\lambda}$ and the multiplicative factor $c\boldsymbol{g}(\boldsymbol{w})$ control how strongly the constraints' violations are penalized.

A desirable property of MMDM is that for small enough learning rates $\eta, \gamma$ and large enough damping parameter $c$, there is a region comprising the vicinity of each constrained minimum such that if the parameters' initialization is in that region and the parameters remain bounded, then the algorithm converges to a constrained minimum (Platt & Barr, 1987). Intuitively, the term $c\boldsymbol{g}(\boldsymbol{w})\nabla_{\boldsymbol{w}}\boldsymbol{g}(\boldsymbol{w})$ enforces a quadratic shape on the optimization search space to the neighbourhood of the solution subspace, and this local behavior is stronger the larger the damping parameter $c$. Thus, the algorithm converges to a minimum if the parameters' initialization is in this locally quadratic region.

## 2.4 GROUP FAIRNESS

In this subsection, we mathematically formalize what *group fairness* means. To ease the exposition, we describe this notion in the central setting, i.e. where the data is not distributed across users.

Consider a dataset $d = (\boldsymbol{z}_1, \boldsymbol{z}_2, \ldots, \boldsymbol{z}_n)$ of $n$ samples. The dataset is partitioned into groups from $\mathcal{A}$ such that each sample $\boldsymbol{z}_i$ belongs to group $a_i$. Group fairness considers how differently a model treats the samples belonging to each group. Many fairness metrics can be written in terms of the similarity of the expected value of a function of interest $f$ of the model evaluated on the general population $d$ with that on the population of each group $d_a = \{\boldsymbol{z}_i \in d : a_i = a\}$ (Agarwal et al., 2018; Fioretto et al., 2020). That is, if we consider a supervised learning problem, where $\boldsymbol{z}_i = (\boldsymbol{x}_i, \boldsymbol{y}_i, a_i)$ and where the output of the model is an approximation $\hat{\boldsymbol{y}}_i$ of $\boldsymbol{y}_i$, we say a model is fair if

$$\mathbb{E}[f(\boldsymbol{X}, \boldsymbol{Y}, \hat{\boldsymbol{Y}}) \mid A = a] = \mathbb{E}[f(\boldsymbol{X}, \boldsymbol{Y}, \hat{\boldsymbol{Y}})] \text{ for all } a \in \mathcal{A}. \tag{4}$$

Most of the group fairness literature focuses on the case of binary classifiers, i.e. $\hat{y}_i \in \{0, 1\}$, and on the binary group case, i.e. $\mathcal{A} = \{0, 1\}$. However, many of the fairness metrics can be extended to general output spaces $\mathcal{Y}$ and categorical groups $\mathcal{A}$. It is common that the function $f$ is the indicator function $\mathbb{1}$ of some logical relationship between the random variables, thus turning (4) to an equality between probabilities. As an example, we describe two common fairness metrics that will be used later in the paper. For a comprehensive survey of different fairness metrics and their inter-relationships, please refer to Verma & Rubin (2018); Castelnovo et al. (2021).

**False negative rate (FNR) parity (or equal opportunity) (Hardt et al., 2016)** This fairness metric is designed for binary classification and binary groups. It was originally defined as equal true positive rate between the groups, which is equivalent to an equal FNR between each group and the overall population. That is, if we let $f(\boldsymbol{X}, Y, \hat{Y}) = \mathbb{E}[\mathbb{1}(\hat{Y} = 0) \mid Y = 1]$ then (4) reduces to

$$\mathbb{P}[\hat{Y} = 0 \mid Y = 1, A = a] = \mathbb{P}[\hat{Y} = 0 \mid Y = 1] \text{ for all } a \in \mathcal{A}. \tag{5}$$

This is usually a good metric when the target variable $Y$ is something positive such as being granted a loan or hired for a job, since we want to minimize the group disparity of misclassification among the individuals deserving such a loan or such a job (Hardt et al., 2016; Castelnovo et al., 2021).

**Accuracy parity (or overall misclassification rate) (Zafar et al., 2017)** This fairness metric is also designed for binary classification and binary groups. Nonetheless, it applies well to general tasks and categorical groups. In this case, if we let $f(\boldsymbol{X}, Y, \hat{Y}) = \mathbb{1}(\hat{Y} = Y)$ then (4) reduces to

$$\mathbb{P}[\hat{Y} = Y \mid A = a] = \mathbb{P}[\hat{Y} = Y] \text{ for all } a \in \mathcal{A}. \tag{6}$$

This is usually a good metric when there is not a clear positive or negative semantic meaning to the target variable, and also when this variable is not binary.

## 3 AN ALGORITHM FOR FAIR AND PRIVATE FEDERATED LEARNING (FPFL)

In this section, we describe the proposed algorithm to enforce fairness in PFL of parametric models learned with SGD. First, we describe an adaptation of the MMDM algorithm to enforce fairness in standard central learning. Then, we extend that algorithm to PFL.

### 3.1 Adapting the MMDM Algorithm to Enforce Fairness

Consider a dataset $d = (\boldsymbol{z}_1, \boldsymbol{z}_2, \ldots, \boldsymbol{z}_n)$ of $n$ samples. The dataset is partitioned into groups from $\mathcal{A}$ such that each sample $\boldsymbol{z}_i$ belongs to group $a_i$. Then, let us consider also a supervised learning setting where $\boldsymbol{z}_i = (\boldsymbol{x}_i, y_i, a_i)$, the model's output $\hat{y}_i$ approximates $y_i$, and the model is parametrized by the parameters $\boldsymbol{w} \in \mathbb{R}^p$. Finally, assume we have no information about the variables' $(\boldsymbol{X}, Y, A)$ distribution apart from the available samples.

We concern ourselves with the task of finding the model's parameters $\boldsymbol{w}^\star$ that minimize a loss function $\ell$ across the data samples and enforce a measure of fairness on the model. That is, we substitute the expected values of the fairness constraints in (4) with empirical averages:

$$\boldsymbol{w}^\star = \underset{\boldsymbol{w} \in \mathbb{R}^p}{\arg\min} \frac{L(d, \boldsymbol{w})}{n} \quad \text{s.t.} \left| \frac{F(d', \boldsymbol{w})}{n'} - \frac{F(d'_a, \boldsymbol{w})}{n'_a} \right| \leq \alpha, \text{ for all } a \in \mathcal{A}, \quad \text{(P2)}$$

where $L(d, \boldsymbol{w}) := \sum_{\boldsymbol{z} \in d} \ell(\boldsymbol{z}, \boldsymbol{w})$, $F(d', \boldsymbol{w}) := \sum_{\boldsymbol{z} \in d'} f(\boldsymbol{z}, \boldsymbol{w})$, $d'$ is a subset of $d$ that varies among different fairness metrics, $n'$ is the number of samples in $d'$, $d'_a$ is the set of samples in $d'$ such that $a_i = a$, $n'_a$ is the number of samples in $d'_a$, $f$ is the function employed for the fairness metric definition, and $\alpha$ is a tolerance threshold. The subset $d'$ is chosen based on the function $f$ used for the fairness metric: if the fairness function does not involve any conditional expectation (e.g. accuracy parity), then $d' = d$; if, on the other hand, the subset involves a conditional expectation, then the $d'$ is the subset of $d$ where that condition holds, e.g. when the fairness metric is FNR parity $d' = \{(\boldsymbol{x}, y, a) \in d : y = 1\}$. We exclude the case where the conditioning in the expectation is $\hat{Y}$, thus excluding predictive parity as a target fairness metric.

Note that the constraints are not strict, meaning that there is a tolerance $\alpha$ for how much the function $f$ can vary between certain groups and the overall population. The reason for this choice is twofold. First, it facilitates the training since the solution subspace is larger. Second, it is known that some fairness metrics, such as FNR parity, are incompatible with DP and non-trivial accuracy. However, if the fairness metric is relaxed, fairness, accuracy, and privacy can coexist (Jagielski et al., 2019; Cummings et al., 2019).

This way, we may re-write (P2) in the form of (P1) to solve the problem with MMDM. To do so we let $\phi(\boldsymbol{w}) = L(d, \boldsymbol{w})/n$ and $\boldsymbol{g}(\boldsymbol{w}) = (g_0(\boldsymbol{w}), g_1(\boldsymbol{w}), \ldots, g_{|\mathcal{A}|-1}(\boldsymbol{w}))$, where

$$g_a(\boldsymbol{w}) = \begin{cases} h_a(\boldsymbol{w}) & \text{if } h_a(\boldsymbol{w}) \geq 0 \\ 0 & \text{otherwise} \end{cases} \quad \text{and} \quad h_a(\boldsymbol{w}) := \left| \frac{F(d', \boldsymbol{w})}{n'} - \frac{F(d'_a, \boldsymbol{w})}{n'_a} \right| - \alpha. \quad (7)$$

Therefore, the parameters are updated according to

$$\begin{cases} \boldsymbol{\lambda} \leftarrow \boldsymbol{\lambda} + \gamma \boldsymbol{g}(\boldsymbol{w}) \\ \boldsymbol{w} \leftarrow \boldsymbol{w} - \eta \left( 1/n \nabla_{\boldsymbol{w}} L(d, \boldsymbol{w}) + \boldsymbol{\lambda}^T \nabla_{\boldsymbol{w}} \boldsymbol{g}(\boldsymbol{w}) + c \boldsymbol{g}(\boldsymbol{w})^T \nabla_{\boldsymbol{w}} \boldsymbol{g}(\boldsymbol{w}) \right) \end{cases}, \quad (8)$$

where we note that $\nabla_{\boldsymbol{w}} \boldsymbol{g}(\boldsymbol{w}) = (\nabla_{\boldsymbol{w}} g_0(\boldsymbol{w}), \nabla_{\boldsymbol{w}} g_1(\boldsymbol{w}), \ldots, \nabla_{\boldsymbol{w}} g_{|\mathcal{A}|-1}(\boldsymbol{w}))$ and

$$\nabla_{\boldsymbol{w}} g_a(\boldsymbol{w}) = \begin{cases} \text{sign}\left( \frac{F(d', \boldsymbol{w})}{n'} - \frac{F(d'_a, \boldsymbol{w})}{n'_a} \right) \left( \frac{\nabla_{\boldsymbol{w}} F(d', \boldsymbol{w})}{n'} - \frac{\nabla_{\boldsymbol{w}} F(d'_a, \boldsymbol{w})}{n'_a} \right) & \text{if } h_a(\boldsymbol{w}) \geq 0 \\ 0 & \text{otherwise} \end{cases}. \quad (9)$$

Now, the fairness-enforcing problem (P2) can be solved with gradient descent/ascent or mini-batch stochastic gradient descent/ascent, where instead of the full dataset $d$, $d'$, and $d'_a$, one considers batches $b$, $b'$, and $b'_a$ (or subsets) of that dataset. Moreover, it can be learned with DP adapting the DP-SGD algorithm from (Abadi et al., 2016), where a clipping bound and Gaussian noise is included in both the network parameters' and multipliers' individual updates. Nonetheless, there are a series of caveats of doing so. First, the batch size $|b'|$ should be large enough to, on average, have enough samples of each group $a \in \mathcal{A}$ so that the difference $\left( \frac{F(b', \boldsymbol{w})}{|b'|} - \frac{F(b'_a, \boldsymbol{w})}{|b'_a|} \right)$ is well estimated. Second, in many situations of interest such as when we want to enforce FNR parity or accuracy parity, the function $f$ employed for the fairness metric is not differentiable and thus $\nabla_{\boldsymbol{w}} F(d', \boldsymbol{w})$ does not exist. To solve this issue, we resort to estimate the gradient $\nabla_{\boldsymbol{w}} F(d', \boldsymbol{w})$ using a differentiable estimation of the function aggregate $F(d', \boldsymbol{w})$; see Appendix A.1 for the details on these estimates in the above situations of interest.

We conclude this section noting the similarities and differences of this work and (Tran et al., 2021). Even though their algorithm is derived from first principles on Lagrangian duality, it is ultimately equivalent to an application of the basic method of differential multipliers (BMDM) to solve a problem equivalent to (P2) when $\alpha = 0$. Nonetheless, the two algorithms differ in three main aspects:

1. BMDM vs MMDM: BMDM is equivalent to MMDM when $c = 0$, that is, when the effect of making the neighbourhood of the solution subspace quadratic is not present. Moreover, the guarantee of achieving a local minimum that respects the constraints does not hold for $c = 0$ unless the problem is simple (e.g. quadratic programming).

2. How they deal with impossibilities in the goal of achieving perfect fairness together with privacy and accuracy. Tran et al. (2021) include a limit $\lambda_{\max}$ to the Lagrange multiplier to avoid floating point errors and reaching trivial solutions, which in our case is taken care by the tolerance $\alpha$, which, in contrast to $\lambda_{\max}$, is an interpretable parameter.

3. This paper uses the exact expression in (9) for the gradient $\nabla_{\boldsymbol{w}}\left|\frac{F(d',\boldsymbol{w})}{n'} - \frac{F(d'_a,\boldsymbol{w})}{n'_a}\right|$, while Tran et al. (2021) ignore the sign of the difference and use $\left|\frac{\nabla_{\boldsymbol{w}}F(d',\boldsymbol{w})}{n'} - \frac{\nabla_{\boldsymbol{w}}F(d'_a,\boldsymbol{w})}{n'_a}\right|$.

In the next subsection, we extend the algorithm to PFL, which introduces two new differences with (Tran et al., 2021). Firstly, the privacy guarantees will be provided for the individuals, and not only to the group to which they belong; and secondly, the algorithm will be tailored to FL.

## 3.2 Extending the Algorithm to Private Federated Learning

In the federated learning setting we now consider that the dataset $d$ is distributed across $K$ users such that each user maintains a local dataset $d^k$ with $n^k$ samples. This setting applies to both cross-device and cross-silo FL, since each sample contains the group information $a_i$. Nonetheless, as in the central setting, the task is to find the model's parameters $\boldsymbol{w}^\star$ that minimize the loss function across the data samples of the users while enforcing a measure of fairness to the model. That is, to solve (P2).

To achieve this goal, we might first combine the ideas from `FederatedSGD` (McMahan et al., 2017) and the previous section to extend the developed adaptation of MMDM to FL. Our adaptation of MMDM cannot be combined with `FederatedAveraging`; for more details see Appendix A.2. In order to perform the model updates from (8), the central server requires the following statistics:

$$\nabla_{\boldsymbol{w}}L(d,\boldsymbol{w}); \quad F(d',\boldsymbol{w}); \quad \left[F(d'_a,\boldsymbol{w})\right]_{a\in\mathcal{A}}; \quad \nabla_{\boldsymbol{w}}F(d',\boldsymbol{w}); \quad \left[\nabla_{\boldsymbol{w}}F(d'_a,\boldsymbol{w})\right]_{a\in\mathcal{A}}; \quad n'; \quad \left[n'_a\right]_{a\in\mathcal{A}}. \tag{10}$$

However, some of these statistics can be obtained from the others: $F(d',\boldsymbol{w}) = \sum_{a\in\mathcal{A}} F(d'_a,\boldsymbol{w})$, $\nabla_{\boldsymbol{w}}F(d',\boldsymbol{w}) = \sum_{a\in\mathcal{A}} \nabla_{\boldsymbol{w}}F(d'_a,\boldsymbol{w})$, and $n' = \sum_{a\in\mathcal{A}} n'_a$. Moreover, as mentioned in the previous section, one might use a sufficiently large batch $b$ of the data instead of the full dataset for each update. Therefore, we consider an iterative algorithm where, at each iteration, the central server samples a cohort of $m$ users $S$ that report a vector with the sufficient statistics for the update, that is

$$\boldsymbol{v}^k = \left[\nabla_{\boldsymbol{w}}L(d^k,\boldsymbol{w}), \left[F((d^k)'_a,\boldsymbol{w})\right]_{a\in\mathcal{A}}, \left[\nabla_{\boldsymbol{w}}F((d^k)'_a,\boldsymbol{w})\right]_{a\in\mathcal{A}}, \left[(n^k)'_a\right]_{a\in\mathcal{A}}\right]. \tag{11}$$

This way, if we define the batch $b$ as the sum of the $m$ users' local datasets $b := \sum_{k\in S} d^k$ and the batch $b'$ analogously, then the aggregation of each user's vectors results in

$$\boldsymbol{v} = \sum_{k\in S} \boldsymbol{v}^k = \left[\nabla_{\boldsymbol{w}}L(b,\boldsymbol{w}), \left[F(b'_a,\boldsymbol{w})\right]_{a\in\mathcal{A}}, \left[\nabla_{\boldsymbol{w}}F(b'_a,\boldsymbol{w})\right]_{a\in\mathcal{A}}, \left[|b'_a|\right]_{a\in\mathcal{A}}\right], \tag{12}$$

which contains all the sufficient statistics for the parameters' update.

Finally, the resulting algorithm, termed Fair PFL or `FPFL` and described in Algorithm 1, inspired by the ideas from (McMahan et al., 2018b; Truex et al., 2019; Granqvist et al., 2020; Bonawitz et al., 2017) guarantees the users' privacy by (i) making sure the aggregation of the users' sent vectors is done securely with secure aggregation (Bonawitz et al., 2017), and (ii) clipping each users' vectors with a clipping bound $C$ and employing the Gaussian mechanism to the sum of the clipped vectors (lines 13 and 15). The variance of the Gaussian noise is calculated according to the refined moments accountant privacy analysis from (Wang et al., 2019), taking into account the number of iterations $T$, the cohort size $m$, the total number of users (or population size) $K$, and the privacy parameters $\epsilon$ and $\delta$. Note that the post-processing property of DP keeps the algorithm private even if the received noisy vector $\boldsymbol{v}$ is processed to extract the relevant information for the model's update.

**Batch size.** This algorithm can also be employed using only a fraction of the user's data in each update, i.e. using a batch $b^k$ of their local dataset $d^k$. Nonetheless, it is desirable (i) to delegate as much computation to the user as possible and (ii) to use as much users' data as possible to have a good approximation of the performance metrics $F(d'_a, \boldsymbol{w})$, which are needed to enforce fairness.

## 4 EXPERIMENTAL RESULTS

We study the performance of the algorithm in two classification tasks. The first task is a binary classification based on some demographic data from the publicly available Adult dataset (Dua & Graff, 2017). The fairness metric considered for this task is FNR parity. The second task is a multi-class classification where there are three different attributes. This task considers accuracy parity as the fairness metric and uses a modification of the publicly available FEMNIST dataset (Caldas et al., 2018), where only digits are considered and the classes are digits written with a black pen on a white sheet, digits written with a blue pen on a white sheet, and digits written with white chalk on a blackboard (see Figure 1).

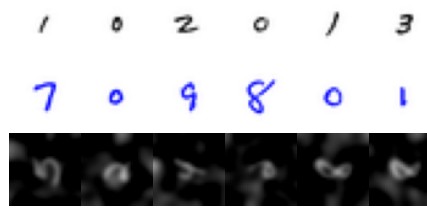

Figure 1: Samples from the Unfair FEMNIST dataset. Digits written on a white sheet with black pen (top), on a white sheet with blue pen (middle), and on a blackboard with white chalk (bottom).

For the first task, we first compare the performance of the MMDM algorithm with BMDM, Tran et al. (2021), and vanilla SGD centrally and without privacy. After that, for both tasks, we confirm how `FederatedSGD` deteriorates the performance of the model for the under-represented classes when clipping and noise (DP) are introduced. Finally, we demonstrate how `FPFL` can, under the appropriate circumstances, level the performance of the model across the groups without largely decreasing the overall performance of the model.

In all our experiments, the fairness metrics are defined as the maximum difference of the value of a performance measure between the general testing data and each of the groups described by that sensitive attribute. Moreover, the privacy parameters are $(\epsilon, \delta) = (2, 1/K)$, where $K$ is the population size. All experimental details and additional experiments are in Appendix B.

### 4.1 RESULTS ON THE ADULT DATASET

We start our experiments comparing the performance and fairness in the central, non-federated, non-private setting. We trained a shallow network (consisting of one hidden layer with 10 hidden units and a ReLU activation function) with these algorithms, where we tried to enforce FNR parity with a tolerance $\alpha = 0.02$. For MMDM, the damping parameter is $c = 2$. For Tran et al. (2021), the multiplier threshold is $\lambda_{\max} = 0.05$. The results after 1,000 iterations are displayed in Table 1, which also shows the gap in other common measures of fairness such as the equalized odds, the demographic parity, or the predictive parity, see e.g. Castelnovo et al. (2021).

Compared to Federated SGD, the MMDM algorithm reduces the FNR gap from 7% to 0.5%, while hardly reducing the accuracy of the model. Similarly, the gap in the equalized odds (EO), which is a stronger fairness notion than the FNR parity, also decreases from around 7% to 3%. Moreover, the demographic parity (DemP) gap, which considers the probability of predicting one or the other target class, also improves. In terms of predictive parity (PP), which uses the precision as the performance function, the MMDM algorithm did not improve the parity among groups. The BMDM algorithm, which lacks a quadratic term of the fairness function, performs even slightly better in the central scenario. However, the algorithm from Tran et al. (2021), an approximation to BMDM as discussed in Section 3.1, gives only a small improvement. Therefore, we continue our analysis to the federated and private settings only with the BMDM and MMDM algorithms.

The second experiment is to study how federated learning, clipping, and DP decrease the performance for the under-represented groups, thus increasing the fairness gap on the different fairness metrics. For that, we trained the same network with `FederatedSGD` and versions of this algorithm where only clipping was performed and where both clipping and DP where included. They were trained with a cohort size of $m = 200$ for $T = 1,000$ iterations and the model with best training cohort accuracy was selected. The privacy parameters are $(\epsilon, \delta) = (2, 5 \cdot 10^{-5})$. The results

Table 1: Performance in the central (i.e. non-federated) setting on the Adult dataset. The fairness metric optimized is FNR parity. The tolerance for BMDM and MMDM is $\alpha = 0.02$.

| Algorithm | Accuracy | FNR gap | EO gap | DemP gap | PP gap |
|---|---|---|---|---|---|
| SGD | 0.857 | 0.071 | 0.071 | 0.113 | 0.016 |
| Tran et al. (2021) without privacy | 0.856 | 0.048 | 0.048 | 0.108 | 0.036 |
| BMDM | 0.857 | 0.001 | 0.030 | 0.094 | 0.046 |
| MMDM (this paper) | 0.856 | 0.005 | 0.028 | 0.091 | 0.063 |

Table 2: Performance of a neural network on the Adult dataset when trained with FL. The fairness metric considered for BMDM-FL and `FPFL` is FNR parity and the tolerance is $\alpha = 0.02$.

| Algorithm | Accuracy | FNR gap | EO gap | DemP gap | PP gap |
|---|---|---|---|---|---|
| FL | 0.851 | 0.121 | 0.121 | 0.122 | 0.037 |
| BMDM-FL | 0.854 | 0.043 | 0.043 | 0.101 | 0.016 |
| Fair FL | 0.855 | 0.036 | 0.039 | 0.108 | 0.033 |
| FL + Clip | 0.844 | 0.169 | 0.169 | 0.129 | 0.051 |
| BMDM-FL + Clip | 0.851 | 0.052 | 0.052 | 0.105 | 0.008 |
| Fair FL + Clip | 0.853 | 0.018 | 0.029 | 0.090 | 0.016 |
| Private FL | 0.847 | 0.148 | 0.148 | 0.126 | 0.041 |
| BMDM-PFL | 0.850 | 0.023 | 0.035 | 0.097 | 0.009 |
| Fair Private FL | 0.851 | 0.001 | 0.027 | 0.087 | 0.042 |

are displayed in Table 2. The network becomes less fair under all the metrics considered, compared with the central setting. Moreover, the introduction of clipping largely increases the unfairness of the models reaching more than a 16% gap in FNR. The addition of DP requires an increase in cohort size from 200 to 1,000, but does not have a larger effect on the unfairness of the models (see Table 5 in Appendix B.1 for the details). These observations are in line with (Bagdasaryan et al., 2019), where they note that the under-represented groups usually have the higher loss gradients and thus clipping affects them more than the majority groups.

After that, we repeated the above experiment with `FPFL` and BMDM with DP. `FPFL` and BMDM with DP converged to a solution faster than `FederatedSGD`, and thus the models were trained for only $T = 250$ iterations. Here the model with the best training cohort accuracy while respecting the fairness condition on the training cohort data was selected. Note that the fairness condition is evaluated with noisy statistics, so a model may be deemed as fair while slightly violating the desired constraints. The results are also included in Table 2 to aid the comparison. We note how, similarly to the central case, the model trained with `FPFL` manages to enforce the fairness constraints while keeping a similar accuracy. In contrast, BMDM seems to not be able to find network weights that respect the fairness conditions, as expected by the less favorable ("less convex") loss surface around these solutions. Moreover, clipping does not seem to affect largely the performance of `FPFL` since it compensates the gradient loss clipping with the fairness enforcement.

## 4.2 RESULTS ON THE UNFAIR FEMNIST DATASET

We start our experiments confirming again the hypothesis and findings from Bagdasaryan et al. (2019) that clipping and DP disproportionately affect under-represented groups. For that, we trained a convolutional network (consisting of 2 convolution layers with kernel of size $5 \times 5$, stride of 2, ReLU activation function, and 32 and 64 filters respectively) with `FederatedSGD` and versions of this algorithm where only clipping was performed and where both clipping and DP were included. For the FEMNIST experiments, the privacy parameters are $(\epsilon, \delta) = (2, 2.5 \cdot 10^{-4})$ and the damping parameter is $c = 20$. They were trained with a cohort size of $m = 100$ for $T = 2,000$ iterations and the model with best training cohort accuracy was selected. The results are displayed in Table 3. Similarly to before, clipping increases the accuracy gap from 13% to almost 17%. In this case, since the number of users $K$ is small, the DP noise is large compared to the users' statistics. Therefore, the

accuracy drops from more than 94% with clipping to 80.7% when DP is also used, and the accuracy gap increases to more than 40%.

The second experiment tests whether `FPFL` can remedy the unfairness without decreasing accuracy too much. We trained the same convolutional network, again for $T = 2,000$ iterations, and selected the model with the best training cohort accuracy that respected the fairness condition on the training cohort. When DP noise is not included, `FPFL` reduces the accuracy gap with `FederatedSGD` by around 9% while keeping the accuracy within 1%. We note how, as before, clipping largely does not affect the ability of `FPFL` to enforce fairness. However, note that since the data is more non-i.i.d. than before (i.e. there are more differences between the distribution of each user) the models that are deemed fair in the training cohort may not be as fair in the general population, and now we see a larger gap between the desired tolerance $\alpha = 0.04$ and the obtained accuracy gap from `FPFL` without noise (0.047 and 0.053 without and with clipping).

Table 3: Performance of a convolutional network on the Unfair FEMNIST when trained with different algorithms: like Table 2, starting with `FederatedSGD` and ending with `FPFL`. The fairness metric considered for `FPFL` is accuracy parity and the tolerance is $\alpha = 0.04$.

| Algo. | Population | Accuracy | Acc. gap |
|---|---|---|---|
| | | $m = 100$ | |
| FL | $K$ | 0.960 | 0.134 |
| FFL | $K$ | 0.950 | 0.047 |
| FL + Clip | $K$ | 0.946 | 0.166 |
| FFL + Clip | $K$ | 0.954 | 0.053 |
| PFL | $K$ | 0.807 | 0.409 |
| FPFL | $K$ | 0.093 | 0.015 |
| | | $m = 2,000$ | |
| PFL | $100K$ | 0.951 | 0.157 |
| FPFL | $100K$ | 0.903 | 0.074 |
| PFL | $1,000K$ | 0.951 | 0.153 |
| FPFL | $1,000K$ | 0.927 | 0.073 |

When DP is used, the noise is too large for `FPFL` to function properly and often the sign of the constraints' gradient, see (9), flips. Note that in the estimation of the performance function, i.e. $F(d_a, \boldsymbol{w})/n_a$, both the numerator and denominator are obtained from a noisy vector, thus making the estimators more sensitive to noise than the estimators for plain `FederatedSGD`. This is due to two main factors: (i) the larger the model, the larger the aggregation of the gradients for each weight, and thus more noise needs to be added; and (ii) the smaller the amount of users, the larger the noise that needs to be added.

For this reason, we considered the hypothetical scenario where the population, used for calculating the DP noise, is 100 and 1,000 times larger, which is a conservative assumption for federated learning deployments (Apple, 2017). Then, we repeated the experiment with DP `FederatedSGD` and `FPFL` and increased the cohort size to $m = 2,000$. In this scenario, DP `FederatedSGD` maintained an accuracy gap of more than 15% while `FPFL` reduced this gap to less than a half in both cases. Nonetheless, the accuracy still decreases slightly, with a reduction of around 5% and 2% respectively compared with DP `FederatedSGD`.

## 5 CONCLUSION

In this paper, we studied and proposed a solution to the problem of group fairness in private federated learning. For this purpose, we adapt the modified method of multipliers (MMDM) (Platt & Barr, 1987) to empirical loss minimization with fairness constraints, producing an algorithm for enforcing fairness in central learning. Then, we extend this algorithm to private federated learning.

Through experiments in the Adult (Dua & Graff, 2017) and a modified version of the FEMNIST (Caldas et al., 2018) datasets, we first confirm previous knowledge that DP disproportionately affects performance on under-represented groups (Bagdasaryan et al., 2019), with the further observation that this is true for many different fairness metrics, and not just for accuracy parity. The proposed `FPFL` algorithm is able to remedy this unfairness even in the presence of DP.

**Limitations and future work.** The `FPFL` algorithm is more sensitive to DP noise than other algorithms for PFL. This requires increasing the cohort size or ensuring that enough users take part in training. Still, for the experiments, the number of users required is still lower (more than an order of magnitude) than the usual amount of users available in deployments of federated learning settings (Apple, 2017). In the future, we intend to scale the algorithm to larger models and empirically integrate it with faster training methods like Federated Averaging (McMahan et al., 2017).

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

# A ADDITIONAL DETAILS ABOUT FPFL

## A.1 DIFFERENTIABLE ESTIMATES OF THE FUNCTION AGGREGATES

As mentioned in Section 3.1, there are many situations of interest when the function $f$ employed to describe the fairness metric is not differentiable. In these cases, the gradient $\nabla_{\boldsymbol{w}} F(d', \boldsymbol{w})$ does not exist and we resort to estimate the gradient $\nabla_{\boldsymbol{w}} F(d', \boldsymbol{w})$ using a differentiable estimation of the function aggregate $F(d', \boldsymbol{w})$. As an example, when trying to enforce FNR parity or accuracy parity (as is the case in our experiments from Section 4 and Appendix B) the employed estimates are:

- Enforcing FNR parity on a neural network $\psi_{\boldsymbol{w}}$ (with a Sigmoid output activation function) in a binary classification task. We note that given an input $\boldsymbol{x}_i$, the raw output of the network $\psi_{\boldsymbol{w}}(\boldsymbol{x}_i)$ is an estimate of the probability that $\hat{y}_i = 1$. Hence, the function aggregate can be estimated as

$$\frac{1}{n'} F(d', \boldsymbol{w}) \approx \frac{1}{n'} \sum_{\boldsymbol{x}_i \in d'} (1 - \psi_{\boldsymbol{w}}(\boldsymbol{x}_i)) \approx \mathbb{P}[\hat{Y} = 0 | Y = 1]. \tag{13}$$

- Enforcing accuracy parity on a neural network $\psi_{\boldsymbol{w}}$ (with softmax output activation function) in a multi-class classification task. We note that given an input $\boldsymbol{x}_i$, the raw $j_{\text{th}}$ output of the network $\psi_{\boldsymbol{w}}(\boldsymbol{x}_i)_j$ is an estimate of the probability that $\hat{y}_i = j$. Hence, if $\boldsymbol{y}_i$ the one-hot encoding vector of $y_i$, the function aggregate can be estimated as

$$\frac{1}{n'} F(d', \boldsymbol{w}) \approx \frac{1}{n'} \sum_{\boldsymbol{x}_i \in d'} \boldsymbol{\psi}_{\boldsymbol{w}}(\boldsymbol{x}_i)^T \boldsymbol{y}_i \approx \mathbb{P}[\hat{Y} = Y]. \tag{14}$$

## A.2 WHY IS FEDERATEDAVERAGING INCOMPATIBLE WITH FPFL?

The proposed algorithm extends the MMDM algorithm from Section 3.1 to FL adapting FederatedSGD, where each user uses all their data, computes the necessary statistics for a model update, and sends them to the third-party.

A natural question could be why is not FederatedAveraging adapted instead. That is, to perform several stochastic gradient descent/ascent updates of the model's parameters $\boldsymbol{w}$ and the Lagrange multipliers $\boldsymbol{\lambda}$, and send the difference of the updated and the original version, i.e. to send the vector $\boldsymbol{v}^k := [\boldsymbol{w}^k - \boldsymbol{w}, \boldsymbol{\lambda}^k - \boldsymbol{\lambda}]$. This way, the size of the communicated vector would be reduced from $(|\mathcal{A}| + 1)p + 2|\mathcal{A}|$ to just $p + |\mathcal{A}|$ and a larger part of the computation would be done locally, increasing the convergence speed. Moreover, the clipping bound could be reduced, thus decreasing the noise necessary for the DP guarantees.

Unfortunately, for the proposed MMDM algorithm, this option could lead to catastrophic effects. Imagine for instance a situation where each user only has data points belonging to one group, say $a = 0$ or $a = 1$. Then in their local dataset the general population is equivalent to the population of their group and thus $F(d^{k'}, \boldsymbol{w}) = F(d_a^{k'}, \boldsymbol{w})$, implying that locally $\boldsymbol{g}(\boldsymbol{w}) = 0$. Therefore, the Lagrange multipliers will never be locally updated and the weights updates will be equivalent to those updates without considering the fairness constraints. That is, using this approach one would (i) recover the same algorithm than standard PFL and (ii) communicate a vector of size $p + |\mathcal{A}|$ instead of size $p$.

## A.3 THE ALGORITHM

The employed algorithm is detailed in algorithm 1. It is separated into three main parts:

1. **Server**: The server initializes the network $\boldsymbol{w}_0$ and Lagrange multiplier $\boldsymbol{\lambda}_0$ parameters, and also calculates the noise scale $\sigma$ using Wang et al. (2019)'s privacy analysis before training. Then, for each iteration $t$ it samples $m$ users $S_t$ and asks for the noisy aggregation of their statistics $\boldsymbol{v}_t$. Then it updates the Lagrange multiplier $\boldsymbol{\lambda}_t$ and the network weights $\boldsymbol{w}_t$ following (8).

2. **Virtual secure third-party (obtained via secure multi-party computation)**: The third party asks for each users' statistics $\boldsymbol{v}^k$ and clips them with the clipping bound $C$. After

that, it aggregates the clipped statistics and privatizes the aggregation with the Gaussian mechanism using the previously calculated by the server noise scale $\sigma$.

3. **Users**: Each user calculates the relevant statistics, see (11), and sends them to the secure virtual third-party.

---

**Algorithm 1:** FPFL. The $K$ users are indexed by $k$, $m$ is the cohort size, $T$ is the number of iterations, $C$ is the clipping bound, and $(\varepsilon, \delta)$ are the DP parameters.

---

**Server Executes:**
> $\boldsymbol{w}_0, \boldsymbol{\lambda}_0 \leftarrow$ InitializeParameters()
> $\sigma \leftarrow$ CalculateNoiseScale$(K, m, \varepsilon, \delta, T)$
> **for** each iteration $t = 1, 2, \ldots, T$ **do**
> > $S_t \leftarrow$ (random set of $m$ users)
> > $\boldsymbol{v}_t \leftarrow$ SecureAggregation$(\boldsymbol{w}_{t-1}, S_t, C, \sigma)$
> > $\boldsymbol{\lambda}_t \leftarrow$ UpdateMultiplier$(\boldsymbol{v}_t, \boldsymbol{w}_{t-1})$
> > $\boldsymbol{w}_t \leftarrow$ UpdateParameters$(\boldsymbol{v}_t, \boldsymbol{\lambda}_t)$
> **end**

**SecureAggregation**$(\boldsymbol{w}, S, C, \sigma)$**:**
> // Performed by the virtual secure third-party
> **for** each user $k$ in $S$ **do**
> > $\boldsymbol{v}^k \leftarrow$ UserStatistics$(k, \boldsymbol{w})$
> > $\boldsymbol{v}^k \leftarrow \boldsymbol{v}^k \cdot \min\left\{1, C/\|\boldsymbol{v}^k\|_2\right\}$
> **end**
> $\boldsymbol{v} \leftarrow \sum_{k \in S} \boldsymbol{v}^k + \mathcal{N}\left(0, C^2 \cdot \sigma^2\right)$
> **return** $\boldsymbol{v}$

**UserStatistics**$(k, \boldsymbol{w})$**:**
> // Performed by the user
> $\boldsymbol{v} \leftarrow \left[\nabla_{\boldsymbol{w}} L(\boldsymbol{w}, d^k), \left[F(\boldsymbol{w}, d'^k_a)\right]_{a \in \mathcal{A}}, \left[\nabla_{\boldsymbol{w}} F(\boldsymbol{w}, d'^k_a)\right]_{a \in \mathcal{A}}, \left[n'^k_a\right]_{a \in \mathcal{A}}\right]$
> **return** $\boldsymbol{v}$

---

## B  EXPERIMENTAL DETAILS AND ADDITIONAL EXPERIMENTS

### B.1  DETAILS OF THE EXPERIMENTS AND ADDITIONAL EXPERIMENTS ON ADULT

#### B.1.1  DETAILS OF THE EXPERIMENTS

**Adult dataset, from the UCI Machine Learning Repository (Dua & Graff, 2017).**  This dataset consists of 32,561 training and 16,281 testing samples of demographic data from the US census. Each datapoint contains various demographic attributes. Though the particular task, predicting individuals' salary ranges, is not itself of interest, this dataset serves as a proxy for tasks with inequalities in the data. A reason this dataset is often used in the literature on ML fairness is that the fraction of individuals in the higher salary range is 30% for the men and only 10% for the women. The experiments in this paper will aim to stop this imbalance from entering into the model by balancing the false negative rate (Castelnovo et al., 2021).

**Federated Adult dataset.**  To generate a version of the Adult dataset suitable for federated learning, it must be partitioned into individual contributions. In the experiments, differential privacy will be guaranteed per contribution. In this paper, the number of datapoints per contribution is Poisson-distributed with mean of 2. Therefore, the number of users is $K \approx n/2 \approx 16,280.5$.

**Privacy and fairness parameters.**  For all our experiments, we considered the privacy parameters $\epsilon = 2$ and $\delta = 5 \cdot 10^{-5} < 1/K$ and the fairness tolerance $\alpha = 0.02$.

**Data pre-processing.**  The 7 categorical variables were one-hot encoded and the 6 numerical variables where normalized with the training mean and variance. There is an underlying assumption

that these means and variances can be learned at a low privacy cost. Hence, to be precise, the private models are $(\epsilon_0 + 2, 5 \cdot 10^{-5})$-DP, where $\epsilon_0$ is a small constant representing the privacy budget for learning said parameters for the normalization.

**Models considered.**  We experimented with two different fully connected networks. The first network, from now on the shallow network, has one hidden layer with 10 hidden units and a ReLU activation function. The second network, henceforth the deep network, has three hidden layers with 16, 8, and 8 hidden units respectively and all with ReLU activation functions. Both networks ended with a fully connected layer to a final output unit with a Sigmoid activation function.

**Hyperparameters.**  For all the experiments, the learning rate was $\eta = 0.1$ for the network parameters and $\gamma = 0.01$ for the Lagrange multipliers. The damping parameter was $c = 2$. The batch size for the experiments learned centrally was $n_{\text{batch}} = 400$ and the cohort sized studied for the federated experiments were $m = 200$ and $m = 1,000$. Finally, the clipping bounds for the shallow and the deep networks were, respectively and depending if the training algorithm was PFL or FPFL, $C = 1.3$ or $C = 2$ and $C = 2$ or $C = 2.4$. These hyper-parameters where not selected with a private hyper-parameter search and were just set as an exemplary configuration. For Tran et al. (2021), we chose the multiplier threshold $\lambda_{\text{max}} = 0.05$ by sweeping over multiple orders of magnitude and finding the optimal accuracy and FNR gap. If one desires to find the best hyper-parameters, one can do so at an additional privacy budget cost following e.g. Abadi et al. (2016, Appendix D), Liu & Talwar (2019).

### B.1.2 ADDITIONAL EXPERIMENTS

We replicated all the experiments with the shallow network from Section 4.1 with the deep network instead. The results for these experiments on the central and federated setting are displayed in Table 4 and Table 5, respectively.

The results obtained with the deep network are almost identical to those of the shallow network in the central setting without DP noise.

The first difference with the results for the shallow network is that the performance and fairness of the deep network does not change much when going from the central to the federated setting without DP noise nor clipping. The shallow network, however, becomes less fair under all the metrics considered.

The results for federated learning with clipping do not differ much between the shallow and deep networks. In both cases we see how `FederatedAveraging` with clipping deteriorates the fairness of the model with all the measures considered. Moreover, they also suggest how `FPFL` can mitigate this problem, maintaining similar levels of accuracy (in this case, even higher) while keeping the fairness below the fairness tolerance.

Another contrast with the shallow network appears in the comparison of algorithms with and without differential privacy. With DP, training does not reliably converge. This is partly due to the fact that the noise is large enough so that sign of the constraints' gradient, see (9), is sometimes mistaken.

For this reason, we repeat the experiments with PFL and `FPFL` with a larger cohort size, $m = 1,000$, to see if a smaller relative noise would aid the training with PFL or with `FPFL`. The results with PFL were almost identical, with similar levels of accuracy and unfairness. On the other hand, the larger signal-to-DP noise ratio helped the models trained with `FPFL` to keep models with the desired levels of FNR gap and lower unfairness measured with any other metric. Moreover, the accuracy of the models, that now work better for the under-represented group, is in fact slightly higher than for the models trained with PFL.

### B.2 DETAILS OF THE EXPERIMENTS ON THE UNFAIR FEMNIST

**FEMNIST dataset (Caldas et al., 2018).**  This dataset is an adaptation of the Extended MNIST dataset (Cohen et al., 2017), which collects more than 800,000 samples of digits and letters distributed across 3,550 users. The task considered is to predict which of the 10 digits or 26 letters (upper or lower case) is depicted in the image, so it is a multi-class classification with 62 possible classes.

Table 4: Performance of a deep and a shallow network on the Adult dataset when trained with SGD and the MMDM algorithm. The fairness metric considered for MMDM is FNR parity and the tolerance is $\alpha = 0.02$. Tran et al. (2021) without privacy, and with $\lambda_{\max} = 0.05$.

| Model | Algorithm | Accuracy | FNR gap | EO gap | DemP gap | PP gap |
|-------|-----------|----------|---------|--------|----------|--------|
| Shallow | SGD | 0.857 | 0.071 | 0.071 | 0.113 | 0.016 |
| Shallow | Tran et al. (2021)* | 0.856 | 0.048 | 0.048 | 0.108 | 0.036 |
| Shallow | BMDM | 0.857 | 0.001 | 0.030 | 0.094 | 0.046 |
| Shallow | MMDM | 0.856 | 0.005 | 0.028 | 0.091 | 0.063 |
| Deep | SGD | 0.858 | 0.070 | 0.070 | 0.117 | 0.006 |
| Deep | Tran et al. (2021)* | 0.853 | 0.054 | 0.054 | 0.111 | 0.035 |
| Deep | BMDM | 0.853 | 0.000 | 0.027 | 0.088 | 0.050 |
| Deep | MMDM | 0.855 | 0.003 | 0.027 | 0.090 | 0.066 |

Table 5: Performance of a deep and a shallow network on the Adult dataset when trained with different algorithms: `FederatedSGD` without privacy, with norm clipping, and with DP, denoted as FL, FL + Clip, and PFL respectively; and `FPFL` without privacy nor norm clipping, with norm clipping only, and with DP, denoted as FFL, FFL + Clip, and FPFL respectively. The fairness metric considered for `FPFL` is FNR parity and the tolerance is $\alpha = 0.02$.

| Model | Algorithm | Accuracy | FNR gap | EO gap | DemP gap | PP gap |
|-------|-----------|----------|---------|--------|----------|--------|
| | | | $m = 200$ | | | |
| Shallow | FL | 0.851 | 0.121 | 0.121 | 0.122 | 0.037 |
| Shallow | BMDM-FL | 0.854 | 0.043 | 0.043 | 0.101 | 0.016 |
| Shallow | FFL | 0.855 | 0.036 | 0.039 | 0.108 | 0.033 |
| Deep | FL | 0.853 | 0.078 | 0.078 | 0.125 | 0.015 |
| Deep | BMDM-FL | 0.850 | 0.051 | 0.051 | 0.116 | 0.012 |
| Deep | FFL | 0.854 | 0.009 | 0.030 | 0.093 | 0.049 |
| Shallow | FL + Clip | 0.844 | 0.169 | 0.169 | 0.129 | 0.051 |
| Shallow | BMDM-FL + Clip | 0.851 | 0.052 | 0.052 | 0.105 | 0.008 |
| Shallow | FFL + Clip | 0.853 | 0.018 | 0.029 | 0.090 | 0.016 |
| Deep | FL + Clip | 0.848 | 0.160 | 0.160 | 0.131 | 0.056 |
| Deep | BMDM-FL + Clip | 0.844 | 0.130 | 0.130 | 0.112 | 0.041 |
| Deep | FFL + Clip | 0.852 | 0.008 | 0.023 | 0.081 | 0.031 |
| Shallow | PFL | 0.828 | 0.171 | 0.171 | 0.093 | 0.038 |
| Shallow | BMDM-PFL | 0.803 | 0.002 | 0.005 | 0.044 | 0.209 |
| Shallow | FPFL | 0.793 | 0.060 | 0.060 | 0.051 | 0.167 |
| Deep | PFL | 0.804 | 0.174 | 0.174 | 0.073 | 0.031 |
| Deep | BMDM-PFL | 0.792 | 0.303 | 0.303 | 0.164 | 0.067 |
| Deep | FPFL | 0.281 | 0.036 | 0.036 | 0.014 | 0.127 |
| | | | $m = 1,000$ | | | |
| Shallow | PFL | 0.847 | 0.148 | 0.148 | 0.126 | 0.041 |
| Shallow | BMDM-PFL | 0.850 | 0.023 | 0.035 | 0.097 | 0.009 |
| Shallow | FPFL | 0.851 | 0.001 | 0.027 | 0.087 | 0.042 |
| Deep | PFL | 0.847 | 0.167 | 0.167 | 0.132 | 0.043 |
| Deep | BMDM-PFL | 0.837 | 0.167 | 0.167 | 0.115 | 0.085 |
| Deep | FPFL | 0.848 | 0.027 | 0.027 | 0.080 | 0.026 |

**Unfair FEMNIST dataset.** We considered the FEMNIST dataset with only the digit samples. This restriction consists of 3,383 users spanning 343,099 training and 39,606 testing samples. The task now is to predict which of the 10 digits is depicted in the image, so it is a multi-class classification with 10 possible classes. Since the dataset does not contain clear sensitive groups, we artificially create three classes (see Figure 1):

- Users that write with a black pen in a white sheet. These users represent the first (lexicographical) 45% of the users, i.e. $\lfloor 0.45 \cdot 3,383 \rfloor = 1,522$ users. These users contain 146,554 (42.7%) training and 16,689 (42.1%) testing samples.

  The images belonging to this group are unchanged.

- Users that write with a blue pen in a white sheet. These users represent the second (lexicographical) 45% of the users, i.e. 1,522 users as well. These users contain 159,902 (46.6%) training and 18,672 (47.1%) testing samples.

  The images belonging to this group are modified making sure that the digit strokes are blue instead of black.

- Users that write with white chalk in a blackboard. These users represent the last remaining 10% of the users, i.e. 339 users. These users contain 36,643 (10.7%) training and 4,245 (10.7%) testing samples.

  The images belonging to this group are modified making sure that the digit strokes are white and the background is black. Moreover, to make the task more unfair, we simulated the blurry effect that chalk leaves in a blackboard. With this purpose, we added Gaussian blurred noise to the image, and then we blended them with further Gaussian blur. To be precise, if $x$ is the image normalized to $[0, 1]$, the blackboard effect is the following.

$$x \leftarrow (x + \xi \circledast \kappa_2) \circledast \kappa_1, \tag{15}$$

  where $\xi_1 \sim \mathcal{N}(0, I)$ is Gaussian noise of the size of the image, $\kappa_1$ and $\kappa_2$ are Gaussian kernels[1] with standard deviation 1 and 2, respectively, and $\circledast$ represents the convolution operation. Moreover, the images are rotated 90 degrees, simulating how the pictures were taken with the device in horizontal mode due to the usual shape of the blackboards.

**Privacy and fairness parameters.** For all our experiments, we considered the privacy parameters $\epsilon = 2$ and $\delta = 2.5 \cdot 10^{-4} < 1/K$. However, for the last experiment, we consider the hypothetical scenario where we had a larger number of users $K \leftarrow 100K$ and $K \leftarrow 1,000K$, thus decreasing the privacy parameter to $\delta \leftarrow \delta/100$ and $\delta \leftarrow \delta/1,000$ and reducing the added noise in the analysis from (Wang et al., 2019).

**Model considered.** We experimented with a network with 2 convolution layers with kernel of size $5 \times 5$, stride of 2, ReLU activation function, and 32 and 64 filters respectively. These layers are followed by a fully connected layer with 100 hidden units and a ReLU activation function, and a fully connected output layer with 10 hidden units and a Softmax activation function. From now on, this model will be referred as the convolutional network.

**Hyperparameters.** For all the experiments the learning rate was $\eta = 0.1$ for the network parameters and $\gamma = 0.05$ for the Lagrange multipliers. The damping parameter was $c = 20$. Note that we set a larger damping parameter to increase the strength to which we want to enforce the constraints, given that the task is harder than before. The cohort sizes considered were $m = 100$ and $m = 2,000$. Finally, the clipping bound for the convolutional network was $C = 250$ if the training algorithm was PFL and $C = 350$ if it was FPFL. As previously, these hyper-parameters where not selected with a private hyper-parameter search and were just set as an exemplary configuration. If one desires to find the best hyper-parameters, one can do so at an additional privacy budget cost following e.g. (Abadi et al., 2016, Appendix D).

---

[1] We used the Gaussian filter implementation from SciPy (Virtanen et al., 2020).

