# OpenReview forum: "Enforcing fairness in private federated learning via the modified method of differential multipliers"
_ICLR.cc/2022/Conference — ICLR 2022 Submitted_

### Official Review · Reviewer_cNjj · 2021-11-01

**Correctness:** 2
**Technical Novelty And Significance:** 3
**Empirical Novelty And Significance:** 3
**Recommendation:** 3
**Confidence:** 4

**Main Review:**

The paper considers a timely and important problem that considers both fairness and privacy; however, the paper has a lot of room for improvement and also the experimental section and results obtained have some flaws and a lot of room for improvement.

Strengths:
1. Paper is considering a timely and important problem.
2. Paper is easy to follow and clear.

Weaknesses:
1. Some important literature review is missing and more importantly these could have been considered as a baseline for comparing their method against methods that are trying to enforce group fairness in FL such as: Cui et al. 2021 Fair and Consistent Federated Learning.
2. Not enough baselines for Fair FL is considered to compare against. It is true that they might not satisfy privacy considerations, but they could still be included for the reader to have intuition of how your method compares against such methods especially that the results are not promising in the paper. Some baselines to consider is Cui et al or Tran et al. Authors claim that they do not compare with Tran et al since Tran et al does not consider privacy, but still I believe these types of results should be included as I am not sure now how effective your approach is in enforcing fairness compared to such baselines.
3. How do authors justify some of the incompatibility results shown in previous work between fairness and privacy?
4. Algorithm 1 can be improved. e.g., What do clients do? What is UpdateMultiplier? What is UpdateParameters? What is Calculate noiseScale?
5. Results are not promising for predictive parity among groups! Discuss why? There is also reduction in accuracy. Perhaps having the baselines that I suggested could improve and give us a sense of what other methods achieve in terms of loss in accuracy etc.
6. In Table 2 put the results for the central setting as well.

**Summary Of The Paper:**

In this paper, authors propose a federated learning algorithm that is able to satisfy group fairness while maintaining privacy.

**Summary Of The Review:**

I think the paper is missing discussion on important previous and related work as well as not including them in the results as baselines. Baselines should be added along with discussion around some of the results which does not seem to be promising. More details can be found under weaknesses discussed above.

---

> ### Author Response · Authors · 2021-11-23
> **Answer to Reviewer cNjj (1)**
>
> We thank the Reviewer for their in-depth and fair review. Please, find below a detailed answer, point by point, to your comments. The answer will be divided into different comments, all titled "Answer to Reviewer cNjj (n)", where n is the order of that comment.
>
> **Weaknesses**
>
> * 1 - *Some important literature review is missing and more importantly these could have been considered as a baseline for comparing their method against methods that are trying to enforce group fairness in FL such as: Cui et al. 2021 Fair and Consistent Federated Learning.*
>
>   We updated our introduction to include the reference to *Agnostic Federated Learning*, *Fair and Consistent Federated Learning*, and *Mitigating Bias in Federated Learning*. The two first methods are methods for group fairness in FL without differential privacy and the last one is a method for group fairness in PFL only for logistic regression.
>
>   Note that still none of these methods address PFL with group fairness for general differentiable models such as neural networks.
>
>   Please, also note that the last two papers, *Mitigating Bias in Federated Learning* and the one suggested by the reviewer (now called *Addressing Algorithmic Disparity and Performance Inconsistency in Federated Learning*), are fairly recent and have not yet presented in any venue. In particular, the one suggested by the reviewer appeared in ArXiv only a month before the conference deadline, so it was hard for us to know about them. Regarding *Agnostic Federated learning*, we forgot to include it, which was our mistake.
>
> * 2 - *Not enough baselines for Fair FL is considered to compare against. It is true that they might not satisfy privacy considerations, but they could still be included for the reader to have intuition of how your method compares against such methods especially that the results are not promising in the paper. Some baselines to consider is Cui et al or Tran et al. Authors claim that they do not compare with Tran et al since Tran et al does not consider privacy, but still I believe these types of results should be included as I am not sure now how effective your approach is in enforcing fairness compared to such baselines.*
>
>   The reviewer suggests we compare with Cui et al.\ 2021 and Tran et al.\ 2020 for fair FL.
>
>   First, as mentioned above, Cui et al.\ 2021 was uploaded to ArXiv a month before the conference deadline, so we did not know about it and, even if we knew, there was not enough time to perform thorough comparisons.
>
>     Second, in the centralized setting, we now compare both with BMDM (the basic method of differential multipliers) and with Tran et al. (2021), but without privacy, since that paper uses a less strong definition than differential privacy. Even so, in our new experiments, the method from Tran et al., which is an approximation to BMDM, gives only small improvements.
>
>   For the federated setting, we have therefore added only the BMDM algorithm and did not extend Tran et al. to FL. As expected, with differential privacy our algorithm, which uses MDMM for a smoother loss function (``more convex" around solutions that comply with the constraints), outperforms BMDM.
>
> * 3 - *How do authors justify some of the incompatibility results shown in previous work between fairness and privacy?*
>
>   Indeed there are trade-offs between fairness, accuracy, and fairness. Some of them are noted in the introduction (2nd paragraph of page 2) and in the reasoning for the proposed solution in the paragraph below (P2).
>
>   There we note that when no perfect fairness is demanded, DP, fairness, and non-trivial accuracy can coexist (see Jagielski et al.,\ 2019 or Cummings et al.,\ 2019). In any case, note how, in Table 3, FPFL breaks for a small cohort size $m = 200$ with an accuracy of $0.093$. That is, it requires a larger cohort size $m=2,000$ to smooth the noise and obtain good accuracy results. After a bugfix, the story is similar for the Adult dataset (see Table 5).

---

> ### Author Response · Authors · 2021-11-23
> **Answer to Reviewer cNjj (2)**
>
> * 4 - *Algorithm 1 can be improved. e.g., What do clients do? What is UpdateMultiplier? What is UpdateParameters? What is Calculate noiseScale?*
>
>   We now delegated Algorithm 1 to Appendix (A.3) in order to accommodate the reviewers' suggestions in the main text. We also used that opportunity to include more preceding text explaining in detail what does each part of the algorithm.
>
>   More precisely, we included a comment below "Secure aggregation" saying "// Performed by the virtual secure third-party" and a comment below "UserStatistics" saying "// Performed by the users". Then, preceding the algorithm we included the following text:
>
>   " The employed algorithm is detailed in Algorithm 1. It is separated into three main parts:
>   1. **Server**: The server initializes the network ${w}_0$ and Lagrange multiplier ${\lambda}_0$ parameters, and also calculates the noise scale $\sigma$ using Wang et al. (2019)'s privacy analysis before training.
>   Then, for each iteration $t$ it samples $m$ users $S_t$ and asks for the noisy aggregation of their statistics ${v}_t$. Then it updates the Lagrange multiplier ${\lambda}_t$ and the network weights ${w}_t$ following~(8).
>
>   2. **Virtual secure third-party (obtained via secure multi-party computation)**: The third party asks for each users' statistics $v^k$ and clips them with the clipping bound $C$. After that, it aggregates the clipped statistics and privatizes the aggregation with the Gaussian mechanism using the previously calculated by the server noise scale $\sigma$.
>
>   3. **Users**: Each user calculates the relevant statistics, see~(13), and sends them to the secure virtual third-party. "
>
> * 5 - *Results are not promising for predictive parity among groups! Discuss why? There is also a reduction in accuracy. Perhaps having the baselines that I suggested could improve and give us a sense of what other methods achieve in terms of loss in accuracy etc.*
>
>   Note that we are optimizing for FNR parity in the Adult dataset and for Accuracy parity in the unfair FEMNIST dataset. These measures of fairness do not have to align, and may even compete with each other in certain situations. This is the reason why when we minimize the disparity for one of these metrics, some of the others may increase.
>
>   The reduction in accuracy is given by the mentioned (by the reviewer and in the paper) trade-off between privacy, fairness, and accuracy.
>
>   In particular, the relevant metrics to observe are the fact that the accuracy is almost maintained while the metric of choice in each case (FNR parity for the Adult dataset and Accuracy parity for Unfair FEMNIST) is reduced and maintained below the pre-defined desired threshold. That is, the proposed solution achieves its objective to train a model with PFL that has a pre-defined level of fairness with respect to a pre-defined fairness metric.
>
> * 6 - *In Table 2 put the results for the central setting as well.*
>
>   The results for the central setting that the reviewer asks for are in Table 1. We presented the results in separate tables to aid our discussion about the obtained results.
>
>   First, we wanted to establish that the MMDM was a method that worked well to enforce fairness while having good accuracy (Table 1).
>
>   Second, we wanted to show we could extend this algorithm to FL and more broadly to PFL (Table 2).

---

### Official Review · Reviewer_FrAm · 2021-11-02

**Correctness:** 4
**Technical Novelty And Significance:** 4
**Empirical Novelty And Significance:** 3
**Recommendation:** 6
**Confidence:** 3

**Main Review:**

Pros:

I think the solution is very elegant as the empirical version of the group fairness (as considered e.g. by Agarwal et al., 2018; Fioretto et al., 2020) can be just added to the DP-SGD training.

The empirical results are convincing: there only a small drop in test accuracy for considerable increase in fairness (as measured by the 'FNR Gap').

The paper is very well written (except for few parts detailed below) :

Cons:

Some weaknesses regarding the presentation (these are fixable I believe) :
-I cannot find there EO Gap, DemP Gap and PP Gap of Tables 1 and 2 explained anywhere. What are those?
-I cannot find any parameters for the DP training in the main text (what was the epsilons / noise level etc.?)

The role of hyperparameter tuning does not seem to be addressed really (at least in the main text). I would be slightly concerned about the fact that the constrained optimisation adds (as far as I see) two hyperparameters (for both SGD and DP-SGD). Especially in DP case this might cause problems (cf. DP hyperparameter tuning, e.g. Liu and Talwar (2019)).

Questions:

-How does the size of the model affect the results? In DP, things get more difficult as the dimension increases, does it affect here?

-There now two additional parameters compared to DP-SGD (for which rigorous hyperparameter tuning already
seems quite difficult). How would you tune the parameters in practice? How sensitive are they?


Other:
This sounds a bit funny:
“where each sample $z_i$ belongs to a group $a_i \in A$. “
perhaps simply “belongs to a group $A$”. ?

-I think that Table 3 would be much more readable if you would write out those abbreviations the way you do in Tables 1 and 2.

- I did not check all the details from the supplementary material, but I think it would be good to add more description of the models (e.g. number of parameters, number of layers etc.) to the main text.


**Summary Of The Paper:**

The fact that DP (and in particular DP-SGD) makes models unfair (e.g. disparate impact on accuracy for minor groups) is well known (Bagdasaryan et al., 2019), however it is possible to have a satisfying trade-off between these ethical measures (Jagielski et al., 2019; Cummings et al., 2019).
I think this seems like an elegant solutions for forcing fairness in DP gradient optimisation trained models.The solution uses classical constrained optimisation (the modified method of differential multipliers) and directly uses an empirical estimate of the group fairness to form the differentiable constraint function.


**Summary Of The Review:**

All in all very nice and elegant looking contribution. However quite a few questions remain, I would be happy to see this paper accepted in case the authors can make those small modifications (that I have listed) and also address my questions.

---

> ### Author Response · Authors · 2021-11-23
> **Answer to Reviewer FrAm**
>
> We thank the Reviewer for their in-depth and fair review. Please, find below a detailed answer, point by point, to your comments.
>
> **Cons**
>
> * *I cannot find there EO Gap, DemP Gap and PP Gap of Tables 1 and 2 explained anywhere. What are those?*
>
> * *I cannot find any parameters for the DP training in the main text (what was the epsilons / noise level etc.?)*
>
>   We now included a parenthesis explaining the name of these fairness metrics after they appear in the text. That is,  Demographic Parity (DemP), Predictive Parity (PP), and Equalized Odds (EO).
>
>   We now also included the important parameters ($\epsilon, \delta$) and some more details of the model architecture in the main text.
>
> * *The role of hyperparameter tuning does not seem to be addressed really (at least in the main text). I would be slightly concerned about the fact that the constrained optimisation adds (as far as I see) two hyperparameters (for both SGD and DP-SGD). Especially in DP case this might cause problems (cf. DP hyperparameter tuning, e.g. Liu and Talwar (2019)).*
>
>   We agree with the reviewer that the problem introduces two new hyper-parameters to the optimization problem. We mention in the appendix that in some settings, optimization of these parameters will take some part of the privacy budget as follows: "These hyper-parameters were not selected with a private hyper-parameter search and were just set as an exemplary configuration. If one desires to find the best hyper-parameters, one can do so at an additional privacy budget cost following e.g. Abadi et al., (2016, Appendix D), Liu \& Talwar (2019)".
>
>   In the setting of federated learning, however, the hyperparameter search is likely to be performed on proxy data in simulation.
>
> **Questions**
>
> * *How does the size of the model affect the results? In DP, things get more difficult as the dimension increases, does it affect here?*
>
>   As the reviewer notes, the larger the model the more difficult it is to train. This is also indicated by our experiments with the convolutional network, which suffered more with PFL and FPFL. Now, we make more emphasis on this fact in Section 4.2 and in the Limitations in the Conclusion.
>
> * *There now two additional parameters compared to DP-SGD (for which rigorous hyperparameter tuning already seems quite difficult). How would you tune the parameters in practice? How sensitive are they?*
>
>   The answer two points above should clarify this one as well.
>
> **Other**
>
> * *This sounds a bit funny: "where each sample $z_i$ belongs to a group $a_i \in \mathcal{A}$. " perhaps simply "belongs to a group $\mathcal{A}$". ?*
>
>   We have modified the text to better explain the link between samples and groups: "Consider a dataset $d = (z_1,z_2, \ldots, z_n)$ of $n$ samples. The dataset is partitioned into groups from $\mathcal{A}$ such that each sample $z_i$ belongs to group $a_i$."
>
> * *I think that Table 3 would be much more readable if you would write out those abbreviations the way you do in Tables 1 and 2.*
>
>   Due to space limitations, we restricted ourselves with the addition of the acronyms after the metrics are written for the first time. Given that they are written close to the tables, we hope it will be clear now.
>
> * *I did not check all the details from the supplementary material, but I think it would be good to add more description of the models (e.g. number of parameters, number of layers etc.) to the main text.*
>
>   As suggested, we included some of the important details of the hyper-parameters now in the main text.

---

> > ### Comment · Reviewer_FrAm · 2021-11-23
> > **Response**
> >
> > I thank the authors for answering basically all my questions. I still wonder about the additional hyperparameters, e.g. how sensitive is the optimisation w.r.t. the choice of these hyperparameters. Notice also that the additional privacy cost of the techniques by Liu and Talwar (2019) that you mention is quite large (and there exist more recent methods, see e.g. arXiv:2110.03620). I will keep my score.

---

> > > ### Author Response · Authors · 2021-11-23
> > > **Hyperparameter tuning**
> > >
> > > We thank the reviewer for their prompt response. Let us elaborate a bit further on hyper-parameter tuning on FL below.
> > >
> > > For actual federated learning deployments (as opposed to central training with DP) it makes the most sense to choose hyper-parameters on proxy data in simulation, and then use those in live training. Hence, the extra privacy budget seen by Liu and Talwar (2019) and the paper mentioned by the reviewer, is usually unnecessary for actual deployments in the industry.
> > >
> > > We did not see results being very sensitive, but to be honest, we did not run sweeps over hyper-parameters, as sections B.1.1 and B.2 mention. The damping parameter $c$ should be set higher the harder the task: in our case, we decided on $c=2$ for the Adult dataset and an order of magnitude larger, $c=20$, for the Unfair FEMNIST dataset (note that it is multiplied by the general learning rate in (8)). Trying 2 or 3 orders of magnitude should be enough for this parameter. It probably makes sense to set the learning rate $\gamma$ for the Lagrange multipliers a bit more conservatively than the general learning rate, so we always set those an order of magnitude smaller. The threshold $\alpha$ is set to obtain some leeway but ensures the model is pretty much fair and therefore does not need to be optimized. In fact, $\alpha$ is directly the fairness objective we set the model to accomplish (at least). We can make this more explicit in sections B.1.1 and B.2 in the camera-ready version.

---

> > > > ### Comment · Reviewer_FrAm · 2021-11-29
> > > > **Reply**
> > > >
> > > > Thanks for clarifying this. I think this basically answers my question. Sounds like the hyperparameters are not that sensitive. The fact that it is enough to try $c$-values for few orders of magnitudes sounds like it is quite stable. I recommend adding these hyperparameter choices your paper. As well as elaborating on the fact that one can avoid the DP cost of hyperparameter tuning in realistic settings.

---

> > > > > ### Author Response · Authors · 2021-11-29
> > > > > **Thanks, we will do that**
> > > > >
> > > > > Thank you for your answer. We will follow your recommendation and add a short discussion on this matter.

---

### Official Review · Reviewer_jvmJ · 2021-11-02

**Correctness:** 2
**Technical Novelty And Significance:** 2
**Empirical Novelty And Significance:** 2
**Recommendation:** 5
**Confidence:** 3

**Main Review:**

- The discussion in the last paragraph of page 2 is confusing. What additional term is considered in this paper?

- The extension to private federated learning seems straightforward. I think the main contribution of the paper is adapting the MMDM algorithm to satisfy fairness constraints. Aggregating $v^k$'s in (11) with clipping and adding Gaussian noise is standard. Similar techniques have been proposed in the federated learning literature.

- The differentiable estimation of $F(d';w) $ is provided for very special cases without any theoretical guarantees on the estimation error in Section 3.1.

- The construction of subsets $d'$ and $d'_a$ is confusing. It seems that these subsets may depend on $w$ in general for example for accuracy parity, which makes the optimization problem in (P2) more difficult.

- Problem (P2) introduces additional hyperparameters such as $c$ and $\gamma$. It is not clear how those parameters are set. In practice, tuning additional hyperparameters is challenging. It is important to understand how sensitive the accuracy and gaps reported in Section 4 are to the choice of additional hyperparameters.

-  The authors introduces cohort size in Section 3.2 without providing much explanation. It will be nice if the authors provide further clarifications.


Minor comments:
Equation (11): what is $k'$? should it be $k$ instead?



**Summary Of The Paper:**

This paper studies the problem of training neural networks with differential privacy and fairness guarantees. The authors proposed an algorithm using a modified method of differential multipliers. In terms of group fairness, the authors focus on false negative rate parity and accuracy parity notions. They compare their proposed algorithm with federatedSGD and its variants with clipping and adding noise.

**Summary Of The Review:**

Several aspects of the proposed method should be clarified, and I think that the extension of the MMDM with fairness constraints to the federate learning is marginally significant.

---

> ### Author Response · Authors · 2021-11-23
> **Answer to Reviewer jvmJ (1)**
>
> We thank the Reviewer for their in-depth and fair review. Please, find below a detailed answer, point by point, to your comments. The answer will be divided into different comments, all titled "Answer to Reviewer jvmJ (n)", where n is the order of that comment.
>
> **Main Review**
>
> * *The discussion in the last paragraph of page 2 is confusing. What additional term is considered in this paper?*
>
>   We agree with the reviewer that "relies on an additional term in the loss function" is imprecise and wrong. We now clarified what we wanted to say with the following: "the method in this paper considers an optimization with fairness constraints, requiring the communication of extra information, and therefore it will extend `FederatedSGD`."
>
> * *The extension to private federated learning seems straightforward. I think the main contribution of the paper is adapting the MMDM algorithm to satisfy fairness constraints. Aggregating $v^k$'s in (11) with clipping and adding Gaussian noise is standard. Similar techniques have been proposed in the federated learning literature.*
>
>   We agree with the reviewer that extending MMDM to FL is one of the main contributions of the paper, which involves deciding on what vector $v^k$ individuals should contribute. We also agree that the privatization of the vector $v^k$, once MMDM is adapted to FL, is not technically difficult. Note that we clearly state that we follow the prior work from (McMahan et al., 2018b; Truex et al., 2019; Granqvist et al., 2020; Bonawitz et al., 2017, and Wang et al., 2019) in the second to last paragraph in page 6.
>
>     Nonetheless, we believe showing that the extension of MMDM to FL to solve a problem with fairness constraints satisfying DP works is not trivial, and it is of independent interest. That is, noting that one may obtain accuracy, fairness, and privacy in FL is also one of the contributions of the paper.
>
> * *The differentiable estimation of $F(d',w)$ is provided for very special cases without any theoretical guarantees on the estimation error in Section 3.1.*
>
>   The presented algorithm works for any differentiable function $F$. In case this is not possible, a differentiable approximation $\hat{F}$ may be used instead. That is the main message we wanted to transmit.
>
>     We showed, for completeness, how we estimated the functions using the output of the neural network as an estimate of the conditional probability. Note that the majority of fairness notions are covered with this approximation, since they are defined in terms of equivalence between probabilities.
>
>     Regarding the theoretical guarantees on the estimation, we followed the standard practice of considering that the output of a neural network trained for classification with a sigmoid activation function in the last layer is the conditional probability $\text{Pr} \lbrace \hat{Y} | X = x \rbrace$. The quality of the estimation depends on two factors: (i) how well the Sigmoid function really estimates the probability of outputting a 0 or a 1 (considering that the prediction is 1 if the output of the Sigmoid function is $\geq 0.5$ and 0 otherwise), and (ii) the number of data instances $x_i$ that we have available to marginalize such conditional probabilities.
>
> * *The construction of subsets $d'$ and $d_a'$ is confusing. It seems that these subsets may depend on in general for example for accuracy parity, which makes the optimization problem in (P2) more difficult.*
>
>   The definition of $d'$ and $d'_a$ does *not* depend on the model $w$ in general. The subset $d'$ depends on the presence or not of a conditional expectation in the definition of the fairness metric $f$.
>
>   For the particular example mentioned by the reviewer, accuracy parity, it is explicitly stated below (P2) that $d' = d$. As mentioned there, the reason is that the fairness function does not involve any conditional expectation.
>
>   In the case where the conditioning variable in the expectation was $\hat{Y}$, then $d'$ would be dependent on the model. We exclude this case, thus excluding the predictive parity metric, which we now clarify at the end of the paragraph below (P2).

---

> ### Author Response · Authors · 2021-11-23
> **Answer to Reviewer jvmJ (2)**
>
> * *Problem (P2) introduces additional hyperparameters such as $c$ and $\gamma$. It is not clear how those parameters are set. In practice, tuning additional hyperparameters is challenging. It is important to understand how sensitive the accuracy and gaps reported in Section 4 are to the choice of additional hyperparameters.*
>
>   We agree with the reviewer that the problem introduces two new hyper-parameters to the optimization problem.  Note we mentioned that in some settings, optimization of these parameters will take some part of the privacy budget, in the appendix, as follows: "These hyper-parameters were not selected with a private hyper-parameter search and were just set as an exemplary configuration. If one desires to find the best hyper-parameters, one can do so at an additional privacy budget cost following e.g. Abadi et al., (2016, Appendix D), Liu \& Talwar (2019)".
>
>     In the setting of federated learning, however, the hyperparameter search is likely to be performed on proxy data in simulation.
>
> * *The authors introduces cohort size in Section 3.2 without providing much explanation. It will be nice if the authors provide further clarifications.*
>
>   We clarify what the cohort is by writing "$\ldots$, at each iteration, the central server samples a cohort of $m$ users $S$ $\ldots$" instead of "$\ldots$, at each iteration, the central server samples $m$ users $S$ $\ldots$" before (11).
>
> **Minor comments**
>
> * *Equation (11): what is $k'$? should it be $k$ instead?*
>
>   Although it seems like it, it is not $k'$, it should be $(d^k)'$, $(d_a^k)'$, and $(n^k_a)'$. We now included such parenthesis to clarify the notation.

---

### Official Review · Reviewer_18Rw · 2021-11-03

**Correctness:** 3
**Technical Novelty And Significance:** 4
**Empirical Novelty And Significance:** 4
**Recommendation:** 3
**Confidence:** 4

**Main Review:**

**Strengths**

- `The paper investigates a relevant problem.` Prior work on fair federated learning has mostly considered achieving uniform performance. In contrast, this work proposes a method for group fairness (in private federated learning), which is a relevant problem (as also empirically evidenced in the paper).
- `The method is novel, technically sound, and achieves its goal.` In contrast to prior work, the proposed approach enforces group fairness constraints via the modified method of differential multipliers. Concretely, it augments the parameter update with a term that encourages the neighborhood of the solution subspace to be quadratic, which guarantees the reachability of a local minimum. The experimental evaluation shows that the proposed method reduces various group fairness gaps while maintaining high accuracy.

**Weaknesses**

- `The method is not compatible with federated averaging.` As explained in section A.2, the proposed method is incompatible with federated averaging, which is quite a big limitation in federated learning. However, this crucially limits the practical applicability of the proposed approach, since federated settings are typically characterized by a communication bottleneck. In contrast to federated SGD, federated averaging avoids this bottleneck by exploiting the availability of local computational resources to learn a global model with significantly fewer communication rounds. I am thus uncertain whether the proposed algorithm would be practically relevant for real-world federated learning applications.
- `The paper does not compare to prior work on group fairness.` While the paper shows that the proposed method can reduce various group fairness metrics in the federated setting, it would be interesting to see how it compares to other methods for group fairness in the centralized setting (e.g., Zafar et al., (2017)). If the method does not outperform prior work in the centralized setting, then it might not be the most promising candidate to consider for the federated setting. An in-depth experimental evaluation of this aspect is definitely needed to motivate the proposed approach. Of course, it may be possible (but unlikely) that prior group fairness methods cannot be extended to the federated setting, but even in that case, the paper should include a detailed discussion to point out the specific limitations.
- `Various aspects of the experimental section are unclear.` See the questions below for some ambiguities. Moreover, section B.1.1 states that delta is approximately 5 * 10^-5, while section B.2 states that delta is approximately 2.5 * 10^-5.
- `The proposed method may be susceptible to a potential privacy violation.` The statistics that need to be shared at every round include the number of individuals per protected attribute for every client. Thus, if the server would set the cohort size to 1 and always select the same user, it would be able to reconstruct the number of individuals per protected attribute for that client (even after the addition of noise when using multiple rounds). This presents a potential privacy leak and could be exploited to learn an unfair model with respect to certain sensitive attributes.
- `Incorrect characterization of prior work.` The paper refers to prior fair federated learning approaches (e.g., Li et al. (2019)) as individual fairness methods. However, these methods enforce uniform performance across all devices, which is neither individual nor group fairness, but a novel notion tailored to the federated learning setting. To view prior work in the individual fairness context, which requires that similar individuals be treated similarly, one would have to consider all clients similar, which beats the purpose of having a similarity notion in the first place. However, I do agree that prior work has not yet studied group fairness in the federated setting (apart from Du et al. (2021), which the paper mentions).

**Questions**

- It is well-known that there is a tradeoff between accuracy and fairness when imposing fairness constraints in machine learning. I believe there is also such a tradeoff in the proposed approach. However, table 2 seems to indicate that fairness can be attained “for free” without losing accuracy (or even increasing accuracy). How can this happen? I suggest conducting more experiments to examine the impact of the various hyperparameters (e.g., the Lagrange multipliers) to better understand this tradeoff.
- Is the accuracy of 9.3% for FPFL with m = 100 in table 3 a typo or does the method break down completely in this case?
- How can you increase K in table 3 to 100K if K denotes the population size?

**Summary Of The Paper:**

This paper considers the problem that differential privacy disproportionally degrades the performance of minority groups in the federated learning context. To mitigate this problem, the paper introduces a novel method for enforcing group fairness constraints via the modified method of differential multipliers. The paper then shows how to generalize this method to the differentially private federated learning setting. Finally, the paper empirically evaluates the method on two real-world datasets, demonstrating that it achieves high accuracy while significantly reducing various group fairness metrics.

**Summary Of The Review:**

While the proposed approach is novel and interesting, various motivational aspects (e.g., comparison with prior work) are unclear or lacking. Moreover, the method is crucially limited by its incompatibility with federated averaging. Thus, the paper does not meet the bar for acceptance in its current form. However, I am willing to increase my score if my concerns are addressed.

---

> ### Author Response · Authors · 2021-11-23
> **Answer to Reviewer 18Rw (1)**
>
> We thank the Reviewer for their in-depth and fair review. Please, find below a detailed answer, point by point, to your comments. The answer will be divided into different comments, all titled "Answer to Reviewer 18Rw(n)", where n is the order of that comment.
>
> **Weaknesses**
>
> * `The method is not compatible with federated averaging.` *As explained in section A.2, the proposed method is incompatible with federated averaging, which is quite a big limitation in federated learning. However, this crucially limits the practical applicability of the proposed approach, since federated settings are typically characterized by a communication bottleneck. In contrast to federated SGD, federated averaging avoids this bottleneck by exploiting the availability of local computational resources to learn a global model with significantly fewer communication rounds. I am thus uncertain whether the proposed algorithm would be practically relevant for real-world federated learning applications.*
>
>   As the reviewer and the paper mention, a limitation in the proposed solution is that it is not compatible with Federated Averaging. This means that the training of the models with FPFL will be slower than with standard PFL. However, they are still trainable with SoTA accuracy (see Tables 2 and 3), so they could be used in practice in some scenarios.
>
>     Extending the algorithm in this paper to Federated Averaging would be valuable, but not at all trivial. In our view, it would require either significant theoretical work, or significant empirical work. It is left as future work.
>
> * `The paper does not compare to prior work on group fairness.` *While the paper shows that the proposed method can reduce various group fairness metrics in the federated setting, it would be interesting to see how it compares to other methods for group fairness in the centralized setting (e.g., Zafar et al., (2017)). If the method does not outperform prior work in the centralized setting, then it might not be the most promising candidate to consider for the federated setting. An in-depth experimental evaluation of this aspect is definitely needed to motivate the proposed approach. Of course, it may be possible (but unlikely) that prior group fairness methods cannot be extended to the federated setting, but even in that case, the paper should include a detailed discussion to point out the specific limitations.*
>
>   In the centralized setting, we now compare both with BMDM (the basic method of differential multipliers) and with Tran et al. (2021), but without privacy, since that paper uses a less strong definition than differential privacy.
>
>   Even so, in our new experiments, the method from Tran et al., which is an approximation to BMDM, gives only a small improvement with respect to SGD.
>
>   For the federated setting, we have therefore added only the BMDM algorithm. As expected, with differential privacy, our algorithm, which uses MDMM for a smoother loss function ("more convex" around solutions that comply with the constraints), outperforms BMDM.
>
> * `Various aspects of the experimental section are unclear.` *See the questions below for some ambiguities. Moreover, section B.1.1 states that delta is approximately 5 * 10^-5, while section B.2 states that delta is approximately 2.5 * 10^-5.*
>
>   The parameter $\delta$ is selected to be approximately $1/K$, where $K$ is the number of users.
>
>   * For the Adult dataset, in Section 4.1, the parameter is $\delta = 5 \cdot 10^{-5} \approx 1/K \approx 6.14 \cdot 10^{-5}$. This results from the fact that (i) there are $n = 32,561$ training data points and the users are selected so that the training data is Poisson distributed with mean 2, and hence $K \approx n/2$.
>
>   * For the FEMNIST dataset, in Section 4.2, the parameter is $\delta = 2.5 \cdot 10^{-4} \approx 1/K \approx 2.96 \cdot 10^{-4}$. This results from the fact that there are $K=3,383$ users.
>
>   We have updated the manuscript to clarify the number of users in the experiments for the Adult dataset. Moreover, we also clarified how the $\delta$ parameter is actually smaller than $1/K$ to favor simpler numbers that benefit the users' privacy.

---

> ### Author Response · Authors · 2021-11-23
> **Answer to Reviewer 18Rw (2)**
>
> * `The proposed method may be susceptible to a potential privacy violation.` *The statistics that need to be shared at every round include the number of individuals per protected attribute for every client. Thus, if the server would set the cohort size to 1 and always select the same user, it would be able to reconstruct the number of individuals per protected attribute for that client (even after the addition of noise when using multiple rounds). This presents a potential privacy leak and could be exploited to learn an unfair model with respect to certain sensitive attributes.*
>
>   Note that the method is not more susceptible to a privacy violation than the DP guarantees that it offers. First, if the cohort size is set to 1, the noise added will be as high as the DP analysis requires, which gives local DP guarantees. Second, an important part of the algorithm (and of the DP analysis) is that at each iteration the server selects $m$ users *at random*. That is, the server cannot target a specific user and sample from them. If that were possible, the DP guarantees for this and almost all current private federated learning approaches would break.
>
> * `Incorrect characterization of prior work.` *The paper refers to prior fair federated learning approaches (e.g., Li et al. (2019)) as individual fairness methods. However, these methods enforce uniform performance across all devices, which is neither individual nor group fairness, but a novel notion tailored to the federated learning setting. To view prior work in the individual fairness context, which requires that similar individuals be treated similarly, one would have to consider all clients similar, which beats the purpose of having a similarity notion in the first place. However, I do agree that prior work has not yet studied group fairness in the federated setting (apart from Du et al. (2021), which the paper mentions).*
>
>   Indeed, the models that aim for similar performance to all devices are not under the category of individual fairness. We did not account for the classical notion of individual fairness from Dwork et al. when we wrote that sentence. We updated the manuscript just writing ``In the realm of FL,  there has been some research studying how to achieve that the performance of the model is similar among devices $\ldots$".
>
> **Questions**
>
> * *It is well-known that there is a tradeoff between accuracy and fairness when imposing fairness constraints in machine learning. I believe there is also such a tradeoff in the proposed approach. However, table 2 seems to indicate that fairness can be attained “for free” without losing accuracy (or even increasing accuracy). How can this happen? I suggest conducting more experiments to examine the impact of the various hyperparameters (e.g., the Lagrange multipliers) to better understand this tradeoff.*
>
> * *Is the accuracy of 9.3% for FPFL with m = 100 in table 3 a typo or does the method break down completely in this case?*
>
>   Indeed there are trade-offs between fairness, accuracy, and fairness. Some of them are noted in the introduction (2nd paragraph of page 2) and in the reasoning for the proposed solution in the paragraph below (P2).
>
>     There we note that when no perfect fairness is demanded, DP, fairness, and non-trivial accuracy can coexist (see (Jagielski et al., 2019; Cummings et al., 2019). In any case, note how, in Table 3, FPFL breaks for a small cohort size $m = 200$ with an accuracy of $0.093$ (not a typo). That is, it requires a larger cohort size $m=2,000$ to smooth the noise and obtain good accuracy results. After a bugfix, the story is similar for the Adult dataset (see Table 5).
>
> * *How can you increase K in table 3 to 100K if K denotes the population size?*
>
>   The population size $K$ is not increased. We consider the *hypothetical* case where we had 100 or 1,000 more population in order to calculate the noise added to the model. Since the data is not available, we could only hypothesize using the available users but calculate the noise as if more users were present. This was more clarified in page 16 of the Appendix and is now further clarified at the end of Section 4.2.

---

### Author Response · Authors · 2021-11-23
**Updated version with new experiments**

Thanks to all four reviewers for their in-depth and fair reviews. We hope we have addressed every one of their concerns in the revised paper. New material and experimental results are in blue in the PDF.

The additional experiments use BMDM, the basic method of differential multipliers, and the method from Tran et al. (2021), an approximation to BMDM. Implementing these exposed a bug in our central training which caused some network weights not to take the additional constraints into account. By fixing this bug, a few numbers changed, but not the overall story. In addition, we found a configuration inconsistency (wrong cohort size) in the federated experiments on the Adult dataset. With this fixed, the experiments with DP require an increased cohort size to converge, consistent with the previous results on FEMNIST. We have triple-checked our other experiments, and feel confident the results are now correct.

---

### Decision · Program_Chairs · 2022-01-20

**Decision:**

Reject

**Comment:**

This paper continues the investigation on fairness and privacy in the context of federated learning. We appreciate the detailed response from the authors. During the rebuttal period, the authors have largely updated the set of experiments, since there was an identified bug in the previous implementation. Another drawback that the AC identified is that there is a lack of formulation and formal guarantees in the paper. In particular, is the proposed algorithm trying to satisfy example-level or client-level data privacy? The resulting noise scale can be quite different. Unlike prior work (e.g. Jagielski et al), the proposed algorithm does not seem to provide any fairness guarantee. Thus, it is not clear why the proposed approach is justified (even under some assumptions). In a similar vein, perhaps the authors could consider a more in-depth discussion that compares their approach with prior work and articulate what advantages does their new method offers. Overall, the paper is not ready for publication at ICLR.